# Spatiotemporal dynamics characterise spectral connectivity profiles of continuous speaking and listening

**Omid Abbasi**[1]*, **Nadine Steingräber**[1], **Nikos Chalas**[1,2], **Daniel S. Kluger**[1,2], **Joachim Gross**[1,2]

**1** Institute for Biomagnetism and Biosignal Analysis, University of Münster, Münster, Germany, **2** Otto-Creutzfeldt-Center for Cognitive and Behavioral Neuroscience, University of Münster, Münster, Germany

* abbasi@wwu.de

**Data Availability Statement:** An example dataset, and the numerical values immediately underlying the Figures, are available at https://osf.io/9fq47/. Raw data, however, are protected by data privacy

## Abstract

Speech production and perception are fundamental processes of human cognition that both rely on intricate processing mechanisms that are still poorly understood. Here, we study these processes by using magnetoencephalography (MEG) to comprehensively map connectivity of regional brain activity within the brain and to the speech envelope during continuous speaking and listening. Our results reveal not only a partly shared neural substrate for both processes but also a dissociation in space, delay, and frequency. Neural activity in motor and frontal areas is coupled to succeeding speech in delta band (1 to 3 Hz), whereas coupling in the theta range follows speech in temporal areas during speaking. Neural connectivity results showed a separation of bottom-up and top-down signalling in distinct frequency bands during speaking. Here, we show that frequency-specific connectivity channels for bottom-up and top-down signalling support continuous speaking and listening. These findings further shed light on the complex interplay between different brain regions involved in speech production and perception.

## Introduction

Turn-taking during a fluent dialogue is often highly efficient and fast. During a natural conversation, the distribution of gaps between turns peaks at 200 ms, which is surprising because producing a single word in a primed picture-naming task takes about 600 ms [1]. Levinson suggests that this efficiency is only possible due to predictive comprehension where speech production is planned before the interlocutor finishes their turn. With a similar reference to prediction, Friston and Frith described human communication as 2 dynamic systems that are coupled via sensory information and operate according to principles of active inference by minimising prediction errors [2]. In this theory of communication, speech production is controlled by the speaker's predictive processing model that allows, for example, adjustments of speech volume, speed, or articulation based on the proprioceptive and auditory feedback. In the listener's brain, the listener's predictive processing model will generate predictions about the timing and content of upcoming speech and these predictions are constantly updated by

laws and cannot be made widely available, but may be made available upon reasonable request (subject to these privacy laws). The code used for this study is available at https://osf.io/9fq47/.

**Funding:** This work was supported by the Interdisciplinary Center for Clinical Research (IZKF) of the medical faculty of Muenster, Grant Number Gro3/001/19 (JG), EFRE, Grant Number EFRE-0400394 (OA), the German Research Foundation, Grant Number GR 2024/5-1; GR 2024/11-1; GR 2024/12 -1 (JG), and KL 3580/1-1 (DSK). The funders had no role in study design, data collection and analysis, decision to publish, or preparation of the manuscript. OA and NS were paid by EFRE-0400394 and GR 2024/5-1. DSK was paid by KL 3580/1-1, GR 2024/12 -1, and Gro3/001/19.

**Competing interests:** The authors have declared that no competing interests exist.

**Abbreviations:** CSD, cross-spectral density; DAI, directed asymmetry index; DFT, discrete Fourier transform; EEG, electroencephalography; EMG, electromyogram; GLM, general linear model; ICA, independent component analysis; IFG, inferior frontal gyrus; LCMV, linearly constrained minimum variance; LSTG, left superior temporal gyrus; MEG, magnetoencephalography; MI, mutual information; SSP, signal space projection; STG, superior temporal gyrus; SVD, singular value decomposition.

and compared to incoming sensory information. This model implies the involvement of predictive processing in both speech perception and speech production, and therefore, a partially shared neural substrate for these 2 cognitive processes [3].

Consistent with this model, previous studies have shown that auditory-evoked brain activity from self-produced speech is reduced compared to passive listening [4–6]. Self-produced speech allows speakers to accurately predict the expected auditory input that is used to suppress the expected sensory consequences of the own motor act (known as sensory attenuation). Interestingly, this suppression is absent when a speaker hears their own voice after its acoustic properties have been changed in real-time [7]. In this case, the acoustic sensory input to the speaker's auditory system deviates from the predicted input. This prediction error is reflected in auditory cortex activity that resembles the activity in response to passively perceived speech.

However, in the human brain, these predictive processes operate at various delays [8,9]: Proprioceptive and other sensory information reach the brain with delays determined by conduction velocities and distance from the brain. Internal processing also requires time, as does the execution of corrective motor commands. These delays might necessitate a discontinuous, rhythmic operation of the predictive processing model. Such a rhythmic mode has been demonstrated in the visuomotor system. For example, slow precision movements are adjusted at a rate of about 6 to 9 Hz from within a cerebello-thalamo-cortical loop [9].

Similarly, a preferred frequency is evident in speech production. Syllables are typically produced at a rate of about 5 per second that coincides with the modulation rate of the speech envelope [10]. This typical rate therefore establishes a communication channel between interlocutors who can expect to receive syllables at this particular rate. This alignment [3] facilitates prediction as it establishes a temporal coupling between the speaker's and the listener's predictive processing operations.

Pertaining to the neural substrates of such models, predictions in the human brain have been related to brain rhythms [11,12]. Brain rhythms are rhythmic fluctuations in neuronal activity that can be recorded noninvasively with electroencephalography (EEG) or magnetoencephalography (MEG) and have been implicated in cognitive processes such as attention, perception, and memory [13,14]. More specifically, studies in the visual domain demonstrate that predictions are mostly communicated in low-frequency bands (alpha, beta), whereas prediction errors are communicated in higher gamma frequencies [15–17].

In the auditory domain, brain rhythms have been studied during continuous speech perception. A consistent finding in these studies is that frequency-specific brain activity as measured with MEG or EEG becomes temporally aligned to the partly rhythmic amplitude variations in continuous speech (e.g., modulation rate). The alignment is thought to be initiated by acoustic edges in the speech waveform (such as onsets or, alternatively, peaks in vowel energy) that lead to a phase resetting of ongoing oscillations in the auditory cortex [18]. As a result of this phase resetting, brain activity will be temporally aligned to the quasi-rhythmic structure in speech. In MEG data, this is reflected in significant cross-correlation or coherence between the speech envelope and brain activity in the auditory cortex [19].

As low-frequency brain rhythms represent cyclic excitability changes in underlying neuronal populations [14,20], stimuli arriving at an optimal phase of an ongoing oscillation (corresponding to high excitability) can be preferentially processed while stimuli arriving at the opposite phase (low excitability) might even remain undetected [21]. In the context of continuous speech processing, the alignment is therefore thought to result in a rhythmic sampling or discretization of the continuous input stream into segments subserving predictions for incoming information and thus further processing. Indeed, temporal alignment appears to be stronger for intelligible and attended speech as previously shown by our group [19] as well as other groups [22–25]. Our previous MEG study [12] could directly confirm a significantly stronger

causal influence of higher-order brain areas (left inferior frontal gyrus and left motor areas) to auditory cortex for intelligible compared to unintelligible speech. Taken together, these studies suggest that brain rhythms orchestrate prediction processes related to the processing of continuous speech.

This relatively new field of speech-brain coupling has almost exclusively focused on speech processing in the listener's brain and largely omitted speech production in the speaker's brain. Therefore, the speech production system generating the continuous speech stimuli used in speech-brain entrainment studies has so far not been integrated into a coherent methodological and conceptual framework. Consequently, we know very little about the role of brain frequency-specific dynamics in speech production and their relationship to quasi-rhythmic components in speech and how it differs from speech perception. Here, we use the experimental and analytical approach of speech-brain coupling to study speech production and directly compare it to speech perception. Specifically, with speech-brain coupling we refer to experimental paradigms that use continuous speech (instead of single words) and analytic techniques that specifically capture temporal alignment of brain activity to rhythmic speech components (e.g., syllable rate). The aim is to unravel the dynamic network of speech production with a focus on how it engages in the quasi-rhythmic components of speech—such as syllable production.

Our study builds on previous research demonstrating that speech production is supported by a distributed network of brain areas including prefrontal, motor, somatosensory, auditory, and associative regions [8,26–29]. Our knowledge about this network relies largely on fMRI [30] and lesion studies [31]. Recently, invasive [28,29,32] and noninvasive [26,33] electrophysiological studies have contributed to the field by capitalising on their superior temporal resolution. However, invasive recordings provide only limited spatial coverage from selected recording sites in patients. MEG/EEG could potentially overcome this limitation but it suffers from movement- and muscle-related contamination of signals. Therefore, noninvasive electrophysiological studies have so far largely focused on single word or syllable production [4,5,34,35] or the preparatory phase of speech production [27,36].

So far, only very few studies have studied the functional link between brain frequency-specific dynamics and speech production. Ruspantini and colleagues [37] instructed participants to perform a syllable repetition task at various frequencies. They found significant coherence between MEG activity from the motor cortex and electromyogram (EMG) activity from lip muscles, peaking at about 2 to 3 Hz. While this study was based on silently voiced speech, Alexandrou and colleagues used spontaneously produced overt speech [38]. They recorded muscle activity with EMG and relied on careful data cleaning that included independent component analysis (ICA) to remove artefacts related to mouth movements. Finally, they analysed the amplitude differences in specific frequency bands between experimental tasks such as speech production of continuous speech versus syllable repetition. This statistical contrast revealed high-frequency band modulation (60 to 90 Hz) in bilateral temporal, frontal, and parietal brain areas. This study nicely demonstrates the feasibility of studying natural speech production with MEG. It also provides further evidence that brain oscillations are sensitive markers of the engagement of brain areas in speech production tasks.

Even fewer studies have specifically investigated speech-brain coupling during continuous speech production. One EEG study nicely demonstrated this temporal coupling in a hyperscanning experiment but did not quantify the effects of speech artefacts on the results and did not identify the relevant brain areas [39]. In a follow-up study, the same group showed that the envelope of self-produced speech was maximally coupled to the speaker's EEG signals at negative lags of 25 ms (i.e., EEG signal preceding the speech envelope) [40]. For listening, maximum coupling was observed at about 100 ms consistent with other studies [19]. Bourguignon

and colleagues studied speech-brain coupling during reading with MEG [41]. They reported significant speech-brain coupling in the speaker at frequencies corresponding to phrases (<1 Hz), words (2 to 4 Hz), and syllables (4 to 8 Hz).

In order to use MEG/EEG approaches for studying speech production, removing the induced speech-related movement artefact is crucial. Recently, we characterised the effects of head movement on the recorded MEG data during speech production [42]. In the same study, we also suggested an artefact rejection approach, based on regression analysis and signal space projection (SSP), to remove the induced head movement artefacts from the recorded MEG data and make the current study feasible.

In summary, the current study was designed to identify brain networks involved in continuous speaking and listening and explore the neural correlates associated with the experimental conditions by addressing the following questions:

First, where and in which frequencies is brain activity significantly coupled to the speech envelope during speaking? Second, where and in which frequencies does brain activity either lead or lag speech envelope during speaking? Third, how is speech-brain coupling different between speaking and listening? Fourth, how is the difference between speaking and listening supported by differential brain connectivity?

We addressed these questions by recording MEG during speaking and listening. We recorded MEG data from 30 participants while they answered 7 questions (60 s each; speaking condition) as well as listened to audio recordings of their own voice from the previous speaking session (listening condition; see Methods for details). We quantified speech-brain coupling using multivariate delayed mutual information (MI) on multitaper spectral estimates that has been shown to have high sensitivity and specificity compared to various other methods [43,44]. We performed our analysis in several sequential steps. First, we describe the brain areas showing significant coupling to the speech envelope during continuous speech together with a spectrum of frequencies that supports this coupling. Next, we dissociate cortical areas where brain activity is significantly coupled to speech envelope with either positive or negative lag, respectively. We then turn to the statistical comparison of speaking and listening conditions to identify similarities and differences in the cortical speech production and perception networks. Our next step was to identify the brain network and directed connectivities within this network that support the cognitive operations in listening and speaking. Hence, we directly tested frequency-specific communication channels for top-down and bottom-up signalling during speech production and perception using multivariate Granger causality. Our speech-brain coupling results separated the temporal areas following the speech envelope (in the theta range) from frontal and motor areas preceding the speech envelope (in the delta range). Moreover, our connectivity results indicated that the feedback signals, connecting higher areas such as motor to superior temporal gyrus (STG), represent predictions and are communicated via slow rhythms (below 40 Hz), whereas feedforward signals (reverse direction) possibly represent prediction errors and are communicated via faster rhythms (above 40 Hz).

## Results

Our first step in the analysis was to identify the cortical networks involved in speech production and perception. We like to note that our description of the network of speech production and listening refers specifically to speech-brain coupling, i.e., we describe the network of areas where neural activity (during speaking or listening) is significantly coupled to the speech envelope. Our approach (except for the analysis of spectral power and Granger causality) is insensitive to other brain areas that are engaged in speaking- or listening-related neural processes that are not synchronous to the speech envelope.

### The speech production network

Our analysis operates on a cortical parcellation with 230 areas [45]. For each parcel, the beam-former-derived time series of all voxels in the parcel and all the 3 source orientations are subjected to singular value decomposition (SVD) and the 3 strongest components are used for multivariate MI analysis [46] with the speech envelope after transformation into the frequency domain (see Fig 1 as well as Methods for details). In other words, neural activity in each parcel is represented by the 3 time series explaining most of the variance and we quantify the degree of synchronisation to the speech envelope with multivariate MI. To capture neural processes that precede or follow speech, we compute MI for 52 equally spaced delays between speech envelope and neural activity ranging from −1 to 1 s and for frequencies between 1 Hz and 10 Hz. Note that we focused on coupling in the frequency range up to 10 Hz because previous studies showed that speech envelope frequencies in this range are important for comprehension [47,48].

Fig 2 shows the statistical contrast of speech-brain coupling during speaking compared to 95th percentile of surrogate data for each parcel and participant. The statistical analysis was conducted for each frequency and delay (and FDR-corrected across these dimensions). The

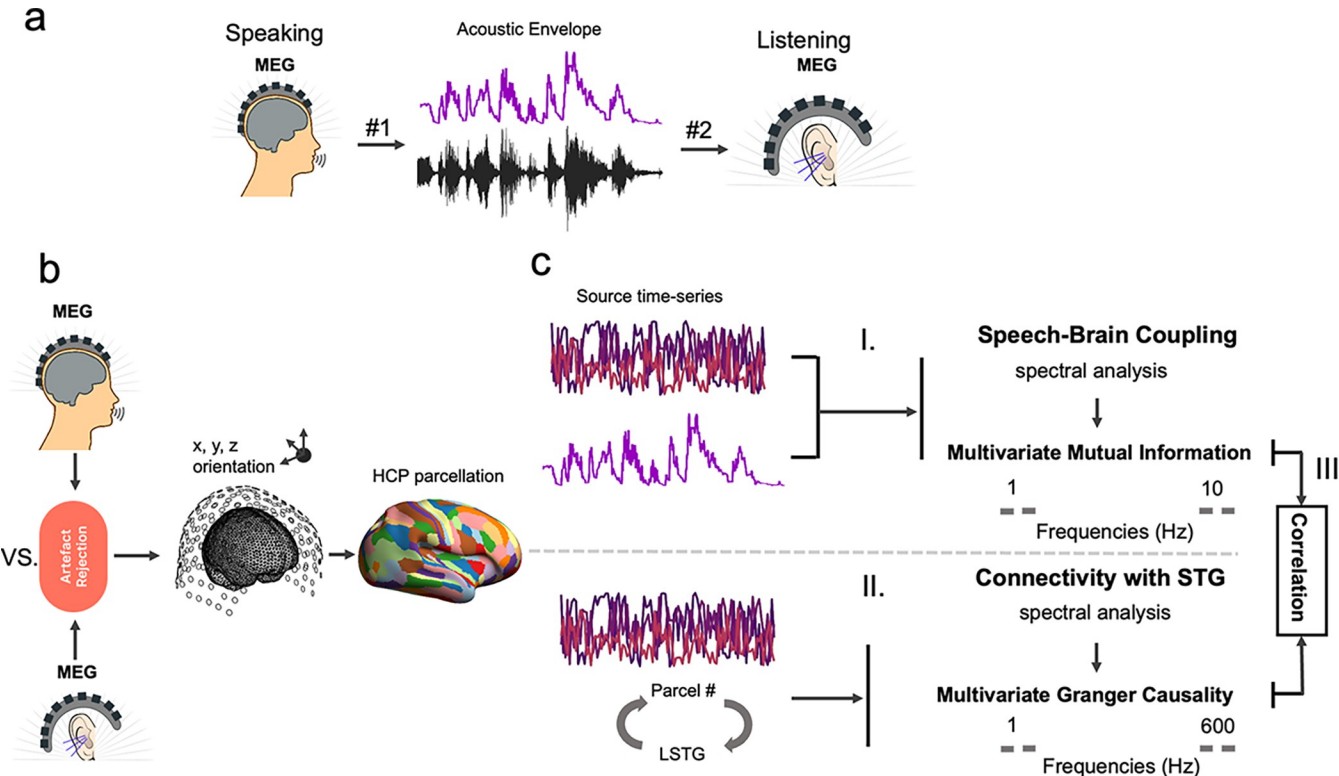

**Fig 1. Methodological pipeline.** (**a**) Participants (*n* = 30) answered to 7 given questions (60 s each; condition #1) as well as listened to audio recordings of their own voice from previous sessions (condition #2) while MEG data was recorded. Amplitude envelope was extracted from the continuous speech signal (see Methods). (**b**) Artefacts were removed from the recorded MEG data [42]. Individual MRIs were used to estimate source models per participant that were interpolated to a template volumetric grid. Cortical areas were divided into 230 anatomical parcels according to the parcellation from the Human Connectome Project [45]. (**c**) For each parcel, estimated source time series were extracted and the first 3 SVD components were subjected to multitaper spectral analysis. We then estimated MI between the speech envelope and complex-valued spectral estimates of all 3 time series (I). Next, using a blockwise approach, we considered the first 3 SVD components of each parcel as a block and estimated the connectivity between the LSTG and other parcels using a multivariate nonparametric Granger causality approach (mCG; II) [49]. Finally, we assessed whether there is a relationship between the directed connectivities and the speech-brain couplings using correlation analysis (III). LSTG, left superior temporal gyrus; MEG, magnetoencephalography; MI, mutual information; SVD, singular value decomposition.

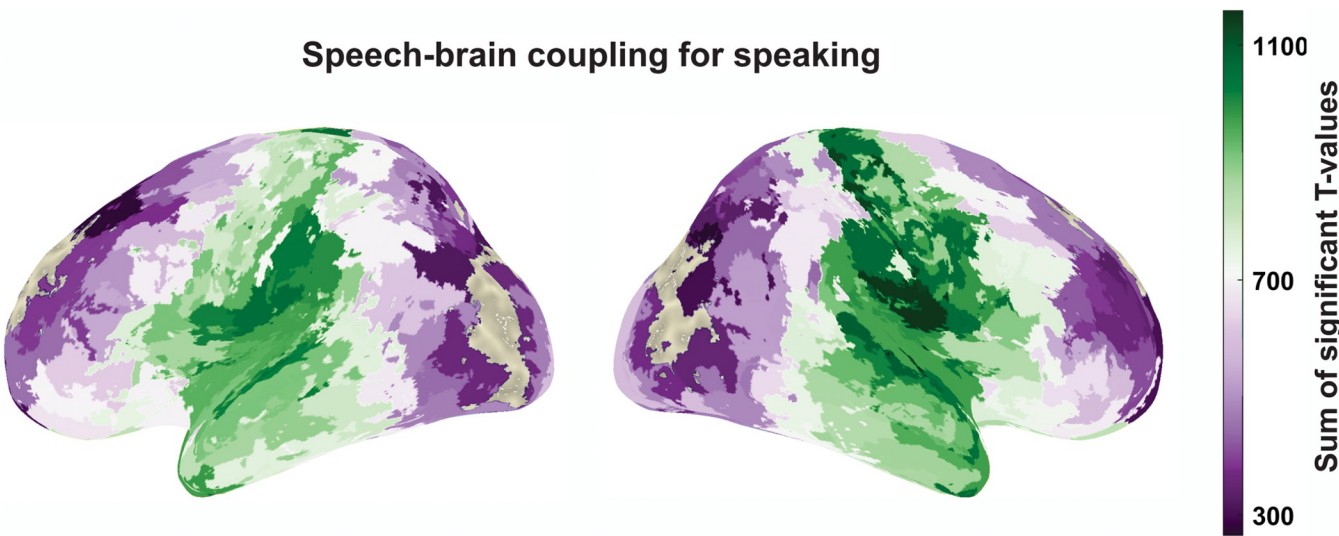

**Fig 2. Speech-brain coupling during speaking.** Cortical map represents groups statistics of multivariate MI between speech envelope and brain activity in each parcel compared to 95th percentile of surrogate data. Colour code represents the sum of all significant T-values across delay and frequency (FDR-corrected across delays (−1 to 1 s), frequencies (1 to 10 Hz), and parcels). The data underlying this figure can be found in https://osf.io/9fq47/. MI, mutual information.

figure shows the sum of significant t-values across time and frequency for each significant parcel. Coupling to the speech envelope can be seen in an extended network of brain areas centred around auditory and motor areas and extending to inferior temporal, inferior frontal, and parietal areas in both hemispheres. Maxima are observed in superior temporal and ventral motor and premotor areas that are known to support speech production [34,43,44] (see S1 Fig for speech-brain coupling during speech perception condition). Before embarking on a more detailed analysis of the dynamic neural pattern that we can resolve with MEG, we aimed to assess the localisation accuracy of our MEG results. Comprehensive comparison with an automated fMRI meta-analysis of speech production networks from neurosynth.org shows very good correspondence with our results. In fact, our speech production network resembled the neurosynth speech production network robustly and significantly more than the neurosynth speech perception network which is surprising given the close resemblance of the speech production and speech perception networks (see S2 Fig).

Next, we aimed to distinguish the brain areas where coupling to the speech envelope is stronger at negative lags (brain activity precedes speech envelope) versus areas where coupling is stronger at positive lags (brain activity follows speech envelope). Negative lags reflect processes of motor/speech planning and motor commands activating articulators, whereas positive lags reflect sensory processing of self-generated speech. The negative lag coupling results beyond the primary motor cortex support the role of the predictive processing model in predicting motor programmes and the sensory consequences of expected speech output. Instead, positive lag coupling may represent prediction error computation that is defined by the mismatch between prior expectation (prediction) and real auditory input.

Fig 3 shows the relative change between summed significant t-values for positive and negative delays (Fig 3A). Purple colours indicate areas with higher t-values at negative compared to positive lags, whereas red colours indicate stronger effects for positive lags. Our results show a clear temporal gradient from the frontal and motor cortex where coupling to the speech envelope is stronger at negative lags preceding speech to temporal and temporoparietal areas following speech.

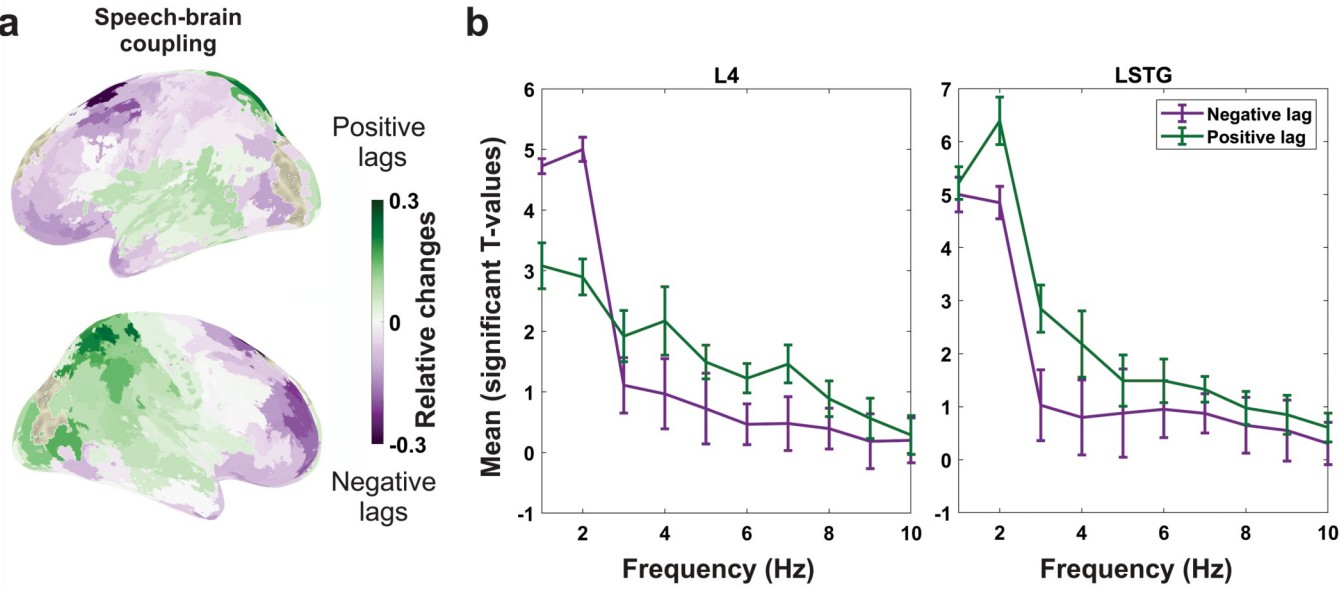

**Fig 3. Speech-brain coupling in different brain areas and frequency bins during speaking.** (**a**) The relative change between summed significant t-values for positive and negative delays. Purple colours indicate areas with higher t-values at negative compared to positive lags, whereas green colours indicate stronger effects for positive lags. (**b**) The spectrum of the averaged significant t-values across positive (green) and negative (purple) delays, for each frequency bin and for L4 and LSTG parcels representing motor and temporal areas, respectively. Error bars represent standard error mean. The data underlying this figure can be found in https://osf.io/9fq47/. LSTG, left superior temporal gyrus.

So far the analysis has focused exclusively on the spatial distribution of brain areas synchronous to the envelope of self-produced speech. However, the speech-brain coupling shown in Fig 3A could be supported by different frequency bands. Therefore, we computed the averaged significant t-values across positive (red) and negative (purple) delays, for each frequency bin, and for L_4 and L_A5 (Fig 3B) parcels representing motor and temporal areas, respectively. The result suggests a dissociation where delta speech-brain coupling dominates at negative lags (synchronous activity preceding speech in frontal and motor areas), while theta speech-brain coupling dominates at positive lags (synchronous activity following speech in parietal and temporal areas).

Next, we aimed to refine the results shown in Fig 3 by mapping the optimal delay of speech-brain coupling for each brain area. This analysis was again based on the statistical map across frequencies and delays. We computed for each area the relative change of speech-brain coupling to the global speech-brain coupling (averaged across all brain areas) for each delay and frequency (see Methods). Fig 4 shows the lag (between −1 s and 1 s) where this relative change averaged across frequencies is maximal. This procedure was selected to enhance local deviations from the general global profile, and it clearly dissociates frontal and motor areas at negative lags from mostly temporal areas at positive lags.

## Speaking versus listening

Having characterised speech-brain coupling for speaking, we comprehensively analysed similarities and differences in speech-brain coupling between speaking and listening conditions. First, to quantify similarities, we computed the correlation of delayed MI (with delays ranging from −1 s to 1 s) for listening and speaking conditions for each parcel and participant. Fig 5A shows group-level cortical maps of these correlations at 2 Hz (represents delta frequency band) and 6 Hz (represents theta frequency band). Across the cortex, the highest similarity

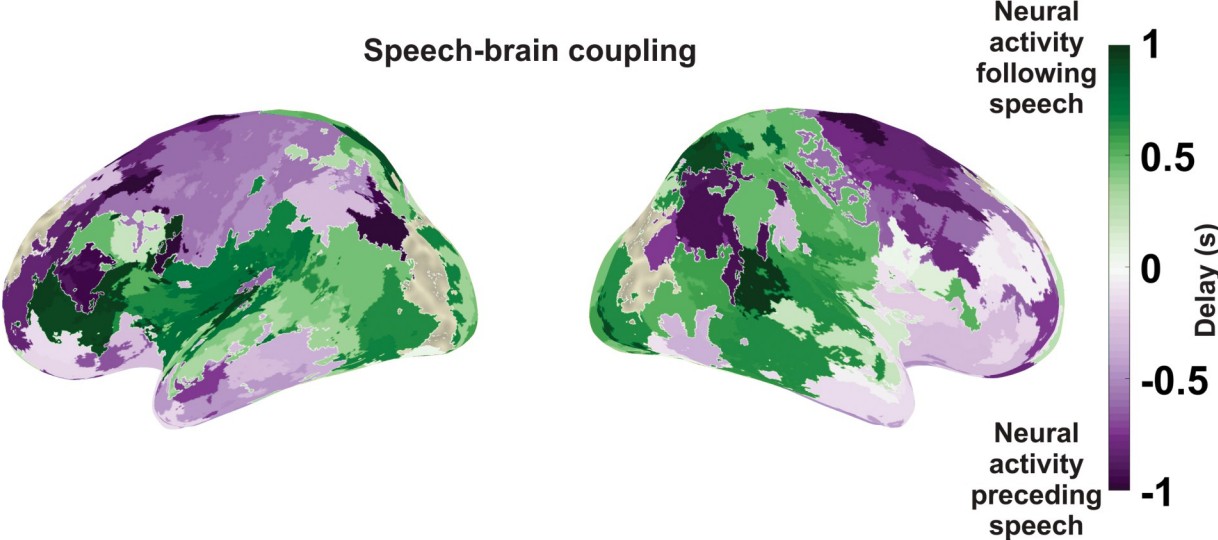

**Fig 4. Speech-brain coupling for different delays during speaking.** The cortical map represents delays between speech envelope and brain activity where the coupling between both is maximal. Purple colours indicate negative delays (brain activity preceding speech) and green colours indicate positive delays (brain activity following speech). The data underlying this figure can be found in https://osf.io/9fq47/.

between conditions in how multivariate speech-brain MI depends on delay can be seen at delta band in superior temporal and inferior motor areas bilaterally. Fig 5B also demonstrates the average of Fisher Z-transformed correlation of lagged MI across all parcels and delays at frequencies from 1 to 10 Hz. These results confirm that the highest similarities between conditions are found at 2 Hz.

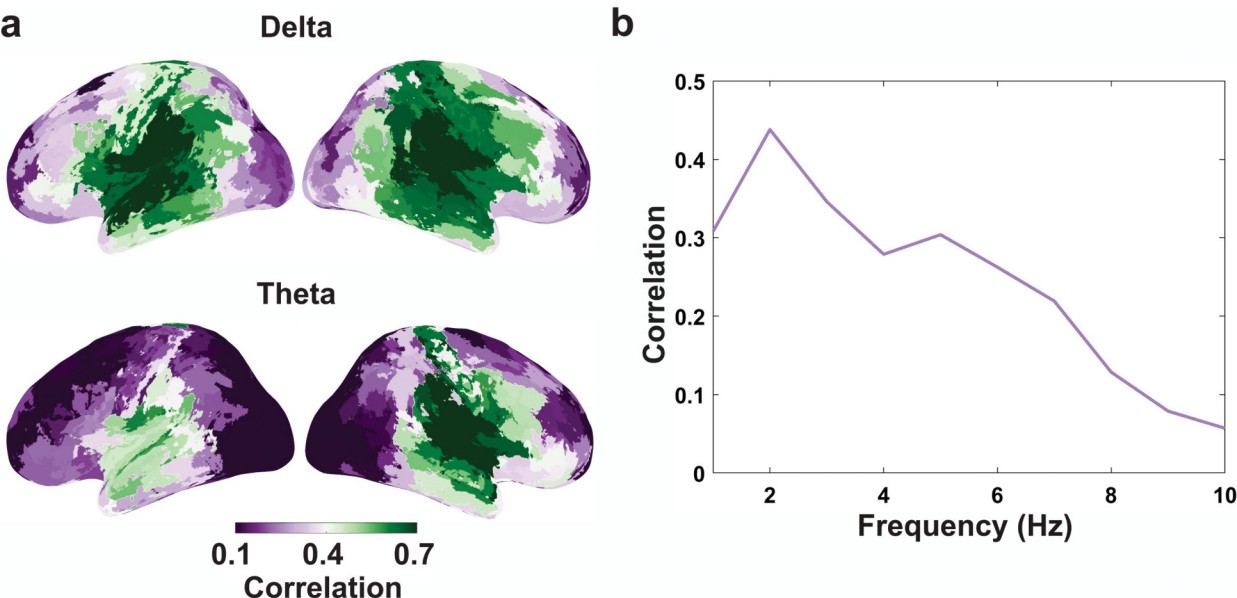

**Fig 5. Comparing speech-brain coupling between speaking and listening conditions.** (**a**) The correlation of lagged MI between speaking and listening for each parcel at delta (2 Hz) and theta (6 Hz). The colorbar illustrates the mean of Fisher Z-transformed correlation across participants. (**b**) The average of Fisher Z-transformed correlation of lagged MI across all parcels and delays at frequencies from 1 to 10 Hz. The data underlying this figure can be found in https://osf.io/9fq47/. MI, mutual information.

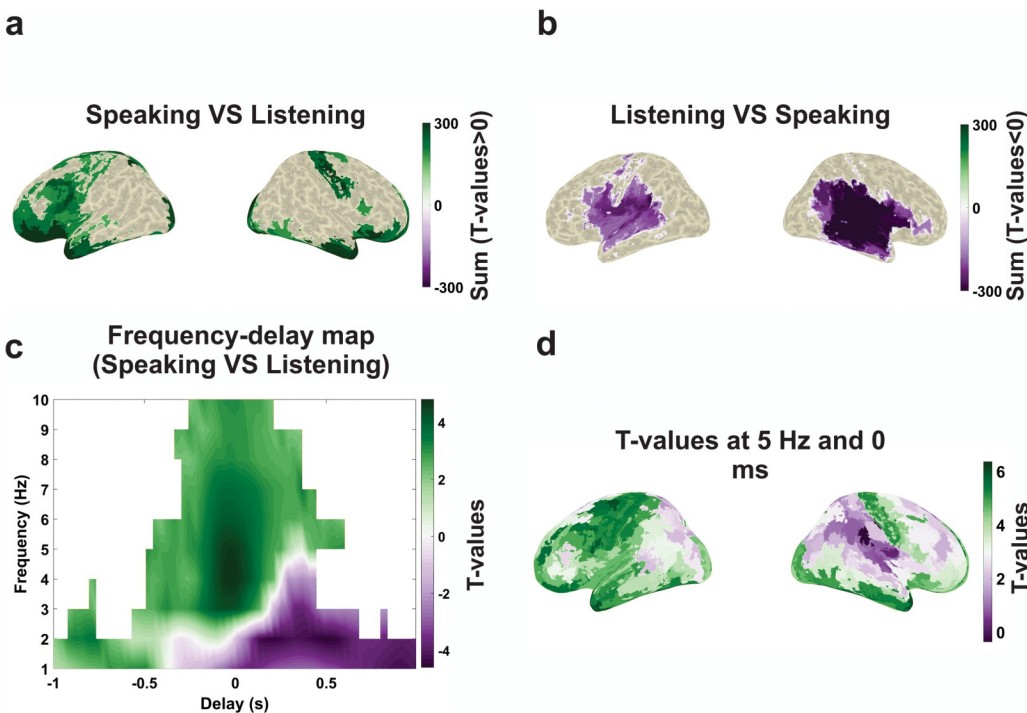

**Fig 6. Statistical comparison of speech-brain coupling in speaking versus listening.** (**a**) Statistical contrast of positive effect (speaking > listening). (**b**) Statistical contrast of negative effect (speaking < listening). (**c**) Frequency-delay profile of statistical contrast averaged over all significant t-values. (**d**) Spatial distribution of t-values at 5 Hz and 0 ms. Note that only the sum of significant t-values was plotted on inflated brain surfaces in a and b ($p < 0.05$). The data underlying this figure can be found in https://osf.io/9fq47/.

Turning to differences between both conditions, we show several results that are based on a single statistical analysis (dependent-samples *t* test of speaking versus listening) across delays (−1 s to 1 s) and frequencies (1 to 10 Hz) while correcting for multiple comparisons across frequencies, delays, and parcels (see Methods). As a result, we obtained t-values for each parcel, frequency, and delay. First, we distinguish between brain areas that show (across all delays) stronger or weaker speech-brain coupling during speaking compared to listening, respectively. Fig 6 shows the sum of significant positive (Fig 6A) and negative (Fig 6B) t-values across all delays and frequencies. Significantly stronger coupling to the speech envelope during speaking was localised within left and right motor areas as well as left inferior frontal and superior temporal areas. We are cautious regarding the interpretation of the orbitofrontal and anterior temporal coupling that is likely due to residual speaking artefacts (such as jaw movements) since this coupling is strongest in atlas parcels closest to the articulators. In contrast, stronger speech-brain coupling during listening compared to speaking was observed in bilateral auditory, inferior frontal, temporal, (pre-) motor, and temporoparietal areas. Comparing panels a and b seems to indicate a left lateralisation to frontal areas stronger for speaking than listening and a right lateralisation in temporal cortex for listening compared to speaking. Therefore, we performed statistical analysis to detect whether there is significant speech-brain coupling lateralisation for both the speaking and listening condition. Our results demonstrate that there is no significant lateralisation for the speaking condition. However, we observed significantly stronger coupling in the right compared to the left temporal area for the listening condition (see S3 Fig). Moreover, both maps (Fig 6A and 6B) show some overlap especially in motor cortex, superior temporal, and inferior frontal areas. This is not a contradiction but merely

indicates the existence of significant negative and positive t-values in the same anatomical area at different delays or frequencies (more detailed comparisons of speech-brain coupling in speaking versus listening at delta (2 Hz) and theta (5 Hz) for different delays (from −600 to 600 ms) can be seen in S4 Fig).

This is indeed supported by Fig 6C that shows the frequency-delay map obtained from averaging all significant t-values across all brain areas. The figure illustrates the spectral-delay profile for speech-brain coupling comparing speaking to listening. In general, speech-brain coupling is stronger for speaking than listening at negative lags (brain activity preceding speech envelope) and stronger for listening than speaking at positive lags. This is expected because speaking requires preparatory predictive processes for predicting motor programmes and the expected sensory consequences of speech occurring at negative lags. Similarly, motor commands from primary motor cortex activating articulators precede the speech envelope during speaking. Also, self-generated speech leads to attenuated auditory processing compared to a passive listening condition [4,5]. This reversal (from positive to negative before and after 0 ms) is prominent in the delta band (1 to 3 Hz). However, the strongest dominance of speaking over listening can be seen in the theta frequency band (around 5 Hz) at a negative lag just before 0 ms. Fig 6D shows the spatial maps of t-values at 5 Hz and 0 ms and reveals left-lateralized dominance of speech-brain coupling in speaking compared to listening in (pre-) motor, left frontal, and temporal brain areas. Since the theta frequency band corresponds to the syllable rate [10], this result indicates that syllable-related activity in these areas is more precisely time-locked to the speech envelope in speaking compared to listening.

## Directed information flow in speaking and listening

Having established the multivariate, frequency-specific speech-brain coupling in speaking and listening, we next investigated the hallmarks of predictive processing models during continuous speaking and listening. Specifically, we tested predictive coding models that have been established in vision and postulate the communication of top-down and bottom-up signals in low- versus high-frequency bands, respectively. To this end, we computed the connectivity of the left superior temporal gyrus (LSTG; depicted in S5 Fig) as a central node in the cortical speech network to all other cortical parcels using multivariate nonparametric Granger causality (mGC) [15,49]. We opted for multivariate Granger causality, first, to be consistent with our multivariate speech-brain coupling analysis and, second, to obtain a more accurate estimate of connectivity by using three-dimensional representations of each parcel activity that capture more signal variance compared to more traditional one-dimensional representations.

Computation of mGC between STG and each cortical parcel resulted in 2 mGC spectra for each pair reflecting both directions (A->B and B->A). Next, we computed the directed asymmetry index (DAI) for each pair of spectra [15]. DAI is the relative difference between both directions and therefore captures the predominant direction of mGC between 2 parcels (see "Methods" for more information). Positive DAI reflects dominant directionality from a parcel towards STG, whereas negative DAI reflects the opposite directionality (from STG to another parcel).

First, we used group statistics to identify brain areas where DAI values in specific frequency bands in the speaking condition differed significantly from zero. We performed group statistics for the following canonical frequency bands: Delta (0 to 4 Hz), theta (4 to 8 Hz), alpha (8 to 12 Hz), beta (12 to 30 Hz), gamma (30 to 60 Hz), and high gamma (60 to 90 Hz). In Fig 7A and 7B, colour codes t-values. Purple colour represents the flow of information from STG to other cortical parcels and the green colour represents the opposite direction.

Clustering statistics revealed significant connectivity during speaking from motor cortex (including precentral gyrus and superior/middle frontal gyrus) to STG in lower frequency

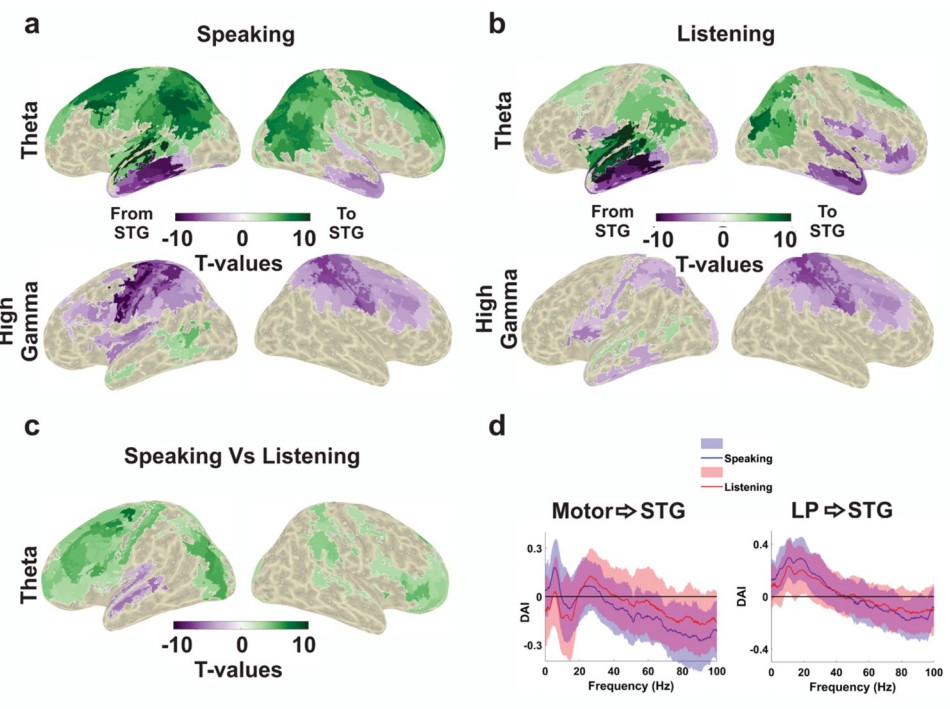

**Fig 7. Connectivity analysis.** (**a**) Significant connectivity between STG and other cortical parcels during speech production in theta and high gamma frequency bands. A cluster-based permutation test was used to detect significant connectivity patterns. Colour codes t-values. Purple colour represents the flow of information from STG to other cortical parcels and the green colour represents the opposite direction. (**b**) Significant connectivity patterns between STG and other cortical parcels during speech perception in theta and high gamma frequency bands. Analysis and colour coding corresponds to panel a. (**c**) Comparison of connectivity patterns between speaking and listening conditions in theta band. A cluster-based permutation test was used to detect significant connectivity differences from all the cortical parcels to STG between speaking and listening conditions. Colour codes t-values. (**d**) Individual spectrally resolved DAI between left motor cortex and LSTG as well as LP and LSTG. Note that only significant values were plotted on inflated brain surfaces ($p < 0.05$). The data underlying this figure can be found in https://osf.io/9fq47/. DAI, directed asymmetry index; LP, left parietal; LSTG, left superior temporal gyrus; STG, superior temporal gyrus.

bands (e.g., theta) and in the opposite direction in high gamma frequencies (Fig 7A). This striking reversal indicates a dissociation of bottom-up and top-down information flow during speaking in distinct frequency bands. Top-down signalling is predominantly communicated in lower frequency bands while bottom-up signalling relies on high-frequency bands (see S6 Fig for all frequency bands).

The listening condition shows an overall similar pattern compared to speaking including the directionality reversal between theta and gamma frequencies (Fig 7B). However, both the top-down effects in the theta band and the bottom-up effects in the gamma band are less pronounced in listening compared to speaking. This is confirmed by a direct comparison of DAI connectivity between speaking and listening.

Fig 7C shows significant connectivity differences between these 2 conditions in the theta band and illustrates significantly stronger top-down signals from frontal/motor cortex to STG in speaking compared to listening and significantly stronger coupling from STG to anterior temporal cortex in listening compared to speaking. Interestingly, our connectivity results illustrate that the top-down effects from different areas to LSTG during speaking occur in distinct low-frequency bands (see S7 Fig for all frequency bands). Fig 7D indicates that the strongest top-down connectivity from the motor cortex to STG occurs at theta (4 to 8 Hz) and beta (12

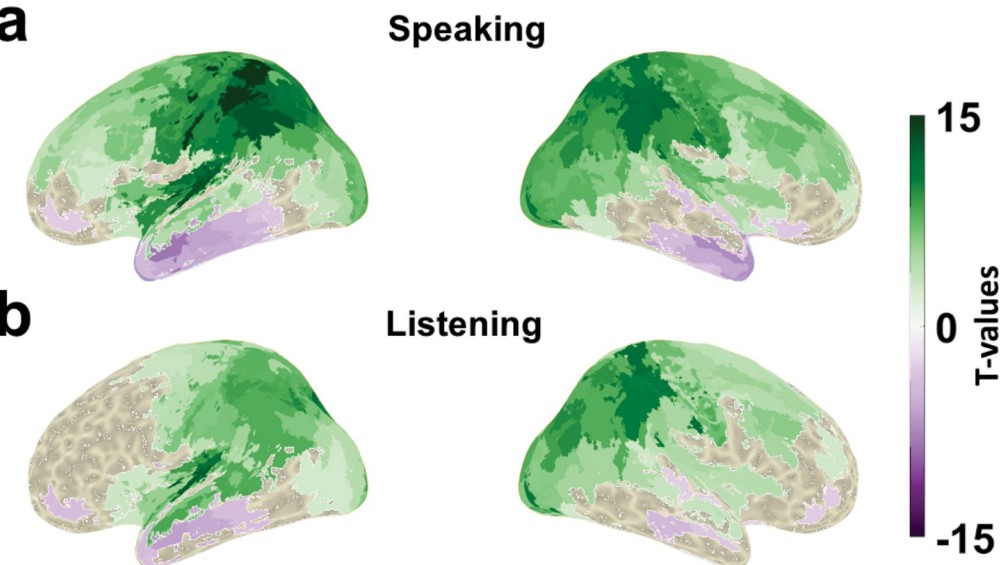

**Fig 8. Low-frequency vs. high-frequency connectivity between all parcels and LSTG.** DAI values in the frequency range 0–30 Hz were compared to DAI values in the frequency range 60–90 Hz in speaking (**a**) and listening (**b**) conditions. Colour codes t-values. Purple colour represents stronger connectivity in high frequencies and green colour represents stronger connectivity in low frequencies. Note that only significant values were plotted on inflated brain surfaces ($p < 0.05$). The data underlying this figure can be found in https://osf.io/9fq47/. DAI, directed asymmetry index; LSTG, left superior temporal gyrus.

to 30 Hz) frequencies, while from the left parietal to STG at alpha frequencies (8 to 13 Hz). Finally, although there is stronger bottom-up connectivity from STG to motor areas in the high gamma band during speaking (compared to listening, Fig 7D), no significant difference was observed.

The general pattern of top-down connectivity in low frequencies and bottom-up connectivity in high frequencies bands is consistent across multiple connections in Fig 7. Therefore, we conducted a dedicated analysis and specifically contrasted the connectivity patterns from cortical parcels to LSTG in low-frequency ranges (0 to 30 Hz) versus high-frequency ranges (60 to 90 Hz). We statistically compared the DAI values of all the cortical parcels in low-frequency ranges versus high-frequency ranges for both the speaking and listening conditions. As we expected according to Fig 7, the motor areas showed the strongest significant difference in the speaking condition, confirming the connectivity from frontal and motor cortex to LSTG in low frequencies and the opposite direction in high frequencies (Fig 8).

Finally, we aimed to connect the 2 main parts of the analysis, namely, the speech-brain and brain-brain connectivities. Specifically, we investigated the relationship between connectivity patterns from all other cortical parcels to STG and the speech-STG coupling. Therefore, we correlated the top-down connectivity indices and MI values (between STG and speech) across participants. We computed correlations separately for each parcel and for different frequency bands for both speaking and listening conditions. This analysis revealed significant negative correlations between top-down beta connectivity from bilateral motor areas as well as left frontal area and the speech-STG coupling in theta band (at 130 ms lag) for speaking condition (Fig 9A; $p < 0.05$). However, this correlation was not observed in the listening condition nor with speech-STG coupling at 0 ms lag (Fig 9B and 9C). Moreover, we conducted the correlation analysis between top-down beta connectivity and speech-STG coupling in each frequency bin in the theta frequency range (at 130 ms lag) that revealed a striking pattern. For low theta

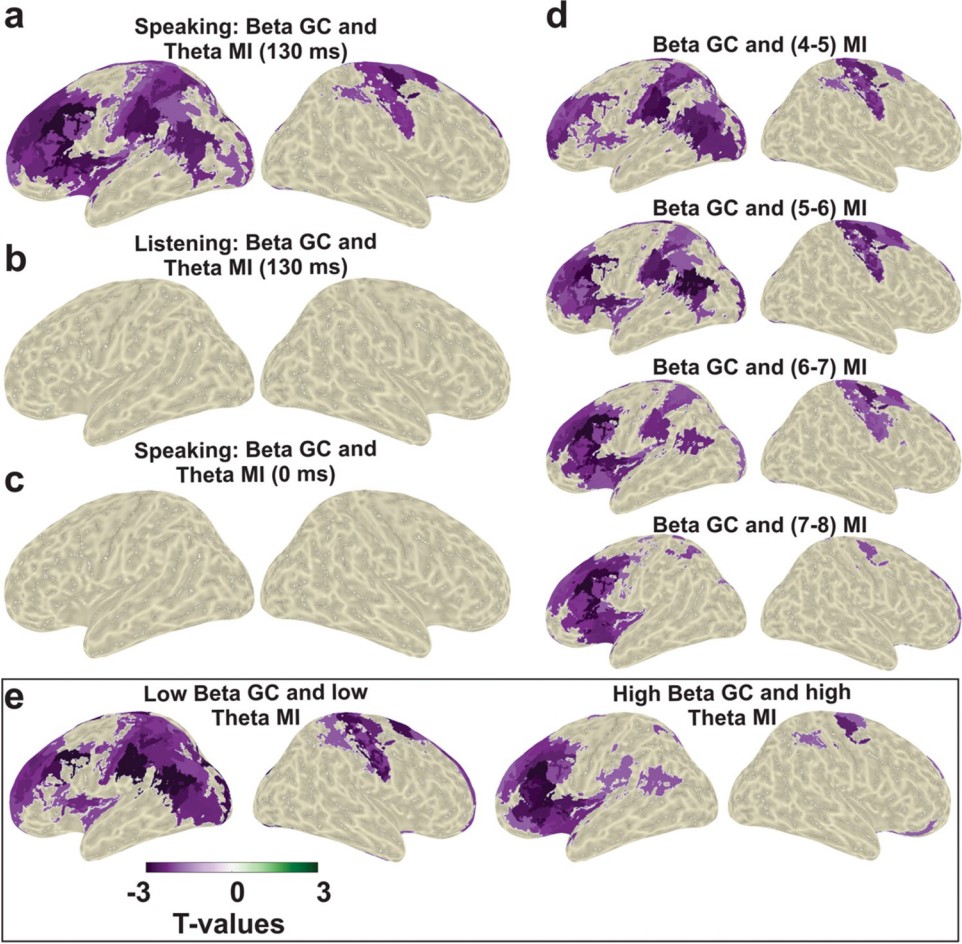

**Fig 9. Correlation analysis between top-down GC and MI.** (**a**) Speech-STG coupling in theta range (positively lagged: 130 ms) is negatively correlated with the top-down beta connectivity from bilateral motor areas and left frontal areas in speaking condition. This effect is largely absent in listening condition (**b**) as well as speaking condition with 0 ms lag (**c**). (**d**) Correlation analysis was conducted between top-down beta connectivity and speech-STG coupling in each frequency bin in theta range. The top-down beta connectivity from bilateral motor areas is negatively correlated with speech-STG coupling in lower theta band (4–6 Hz: first and second rows). However, this correlation was observed between top-down beta connectivity from left frontal area and speech-STG coupling in high theta band (6–8 Hz: third and fourth rows). (**e**) Top-down low beta (12–20 Hz) connectivity originating from bilateral motor areas are negatively correlated with low theta speech-STG coupling. However, top-down high beta (20–30 Hz) connectivity originating from the left frontal area is negatively correlated with speech-STG coupling in the high theta band. The data underlying this figure can be found in https://osf.io/9fq47/. MI, mutual information; STG, superior temporal gyrus.

frequencies, the negative correlation of top-down beta connectivity with speech-STG coupling is significant in parietal and motor areas (Fig 9D). As frequency in the theta band increases, this effect shifts to frontal areas. This transition is most evident in inferior and middle frontal areas where significant clusters are absent in low theta frequencies but present in high theta frequencies (while a superior frontal cluster is largely independent of frequency).

Finally, we further divided the top-down connectivity into low and high beta bands to test which frequency range would be more involved in motor–auditory area interaction during the speaking condition. The main intention of us separating these 2 frequency channels was to see if there is a distinct frequency channel connecting the motor to auditory areas in speech production. We observed that top-down low beta (12 to 20 Hz) connectivity originating from bilateral motor areas are negatively correlated with low theta speech-STG coupling (Fig 9E, left

panel). However, top-down beta (20 to 30 Hz) connectivity originating from the left frontal area is negatively correlated with speech-STG coupling in the high theta band (Fig 9E, right panel). These results further support the importance of beta band activity and connectivity in the motor cortex [50,51]. Due to previous results in the literature, our primary hypothesis was related to the beta band. However, an exploratory analysis across frequency bands revealed significant negative correlation between speech-STG coupling in theta range and top-down delta as well as theta connectivity from mainly left occipital and parietal areas (see S8 Fig).

## Discussion

In this study, we aimed to identify the neural correlates of predictive processes during continuous speaking and listening. Since participants listened to their own previously produced speech, they were likely able to predict some aspects of its content (although both measurements were separated by several days). Still, we opted for this design because it was the only way to guarantee that the same stimulus material was used in both conditions such that differences in both conditions could not have been caused by differences in low-level stimulus features. First, we quantified speech-brain coupling using multivariate delayed MI on multitaper spectral estimates—a method that was recently demonstrated to perform well on simulated and real data [43]. Our findings revealed significant coupling between an extended network of brain areas centred around auditory areas and the speech envelope during continuous speech. These coupling results dissociate frontal and motor areas preceding the speech temporal envelope (in the delta range) from mostly temporal areas following the speech envelope (in the theta range). Furthermore, significantly stronger theta-range speech-brain coupling was observed for speaking (compared to listening) in left and right motor areas and left inferior frontal and superior temporal areas. In contrast, stronger delta-range speech-brain coupling during listening (compared to speaking) was observed in bilateral auditory, inferior frontal, temporal, (pre-) motor, and temporoparietal areas. Moreover, we investigated the directed coupling using multivariate Granger causality within the involved area during speech production and perception. The results of directed coupling analysis indicate significant connectivity from the motor area to STG in lower frequency bands (up to beta) during speaking and in the opposite direction in high gamma frequencies. Notably, the detected beta-range top-down connectivity in the speaking condition was negatively correlated with the speech-brain coupling within the LSTG in the theta band.

### Speech-brain coupling in speaking and listening

Speech-brain coupling represents the alignment of quasi-rhythmic speech components with rhythmic modulations of cortical neural activity that facilitate speech processing [19]. Since previous studies demonstrated that speech envelope frequencies below 10 Hz are important for comprehension, in the first part of our study, we focused on coupling in the frequency range up to 10 Hz [47,48]. We observed significant speech-brain coupling in frequencies below 10 Hz revealing the contribution of a network of brain areas centred on bilateral auditory areas in both speech production and speech perception conditions. This coupling was previously reported in several studies [19,38,41,52,53]. The detected speech-brain coupling was observed bilaterally in both speech production and perception. This finding is in agreement with previous studies indicating bilateral cortical areas are involved in speech perception [54]. Ghinst and colleagues revealed that both the LSTG and right supratemporal auditory cortex tracked speech envelopes at different frequencies [55]. Another study by Gross and colleagues suggested that theta phase entrainment to speech envelope was significant in bilateral auditory areas [19]. The same pattern of activation was observed in Alexandrou and colleagues' report

on speech production and perception. Specifically, they found that power modulation occurs in the frontocentral, parietal, and temporoparietal areas during speech production. However, speech perception is primarily mediated by the temporoparietal and temporal cortices with a pronounced right hemispheric component [38]. Contrary to previous studies, they demonstrated that speech perception and production share cortical representations mainly in the right hemisphere instead of just the left. Interestingly, we also found that bilaterally, inferior and superior temporal motor areas of both conditions shared the largest similarity between them. This clearly indicates that these areas engage in speaking and listening in a similar manner. Moreover, the stronger similarity in the right hemisphere could support the hypothesis that the right hemisphere processes speech meaning in both modalities [38].

Additionally, we found that underlying neural activity during speech production differed significantly from that during speech perception. We observed stronger coupling between left and right motor areas as well as left inferior frontal and superior temporal areas and speech envelope during speaking compared to listening. These results are unsurprising, as speech production requires fine voluntary control of phonation and articulation [37,56]. The beta-band power decrease in widespread bilateral motor areas during continuous speech, presented in our study (see S9 Fig) as well as earlier studies [38,56,57], also indicates that motor areas are actively involved in speech production. A further observation we made was that the speech-brain coupling in the motor areas is stronger at negative lags indicating synchronous activity preceding speech during speaking condition. This coupling at negative lags most likely suggests the involvement of motor preparations or the representation of motor signals involved in controlling speech articulators. Another explanation for the observed negative lags coupling in the motor area could be due to the processing of auditory feedback [29]. Furthermore, the detection of negative lag coupling extending beyond the primary motor cortex lends support to the notion of predicting the sensory consequences of expected speech output before they are actually generated [41,58–60]. Furthermore, another recent study demonstrated that speech-brain coupling increased when reading aloud as compared to listening, providing further evidence that the brain generates sensorimotor representations of self-produced speech [41]. As a result of this internal monitoring system, cortical responses in the auditory cortex are attenuated because speakers can accurately predict the sensory consequences of what they say before they utter it [4–6]. Therefore, a smaller cortical response may be associated with weaker speech-brain coupling in the auditory cortex. The results of our experiments support this hypothesis, as speech-brain coupling in the listening condition was stronger than in the speaking condition in the bilateral auditory cortices.

## Directed information flow in speaking and listening

Our multivariate connectivity analysis provides the first confirmation of frequency-specific communication channels for top-down and bottom-up signalling during continuous speaking and listening in noninvasive recordings of healthy participants. Our results suggest that the implementation of predictive coding via distinct frequency channels that has been demonstrated in the visual domain largely generalises to both continuous speaking and listening in the auditory-motor domain. The model posits that signalling along cortical hierarchies reflect distinct processes: Feedback signals represent predictions and are communicated via slow rhythms (below 40 Hz), whereas feedforward signals represent prediction errors and are communicated via faster rhythms (above 40 Hz) [13,15–17,61].

Only few studies have specifically tested this model in the auditory domain and none during continuous speaking. Previously, highly interesting early work by Fontolan and colleagues demonstrated in invasive recordings distinct frequency channels for feedforward and feedback

communication between 2 hierarchically different auditory areas in 3 epilepsy patients [62]. Also using invasive recordings of epilepsy patients listening to tones, Sedley and colleagues showed that prediction errors are represented in gamma power while predictions are represented in lower frequency beta power [63]. Similar results were obtained in an ECoG study in monkeys [64].

Our multivariate connectivity results significantly extend these reports to noninvasive recordings with whole-brain coverage in our healthy participants that each engage in a listening and a speaking task. Our focus on directed connectivity to and from LSTG, a central node in the auditory hierarchy for extracting meaningful linguistic information from speech input [65], allows the investigation of feedforward and feedback processing during speaking and listening. Indeed, during speaking and listening, multivariate Granger causality from frontal, parietal, and motor cortices to LSTG was significantly stronger in frequencies below 30 Hz, while the opposite direction is significantly stronger in frequencies above 30 Hz. This striking reversal of directionality from low to high frequency bands provides support in the auditory domain regarding the results established in the visual domain mentioned above—namely, the selective communication of feedforward and feedback information in frequency-specific channels. These findings suggest a potential alignment with the "predictive coding" framework that would interpret the top-down signals as predictions and bottom-up signals as prediction errors. Nevertheless, it is important to proceed with caution before drawing definitive conclusions since recent studies have raised concerns about the idea that bottom-up and top-down signals are communicated via distinct frequency channels [66–68]. Therefore, further investigation is needed to fully establish the extent of compatibility between our results and the predictive coding framework. Moreover, further research is needed to clarify if the spatial resolution of MEG with naturalistic tasks as ours is sufficient to reconstruct more detailed functional hierarchies in the auditory domain in a way consistent with previous demonstrations in the visual domain based on strong, high-contrast stimuli [17].

It is however interesting to note that top-down effects from different areas to LSTG during speaking occur in distinct low-frequency bands (Figs 7D and S6). Motor cortex shows strongest top-down signals at theta (4 to 8 Hz) and beta (12 to 30 Hz) frequencies, left inferior frontal cortex at delta (<4 Hz), and left parietal cortex at alpha frequencies (8 to 13 Hz). Theta band connectivity from the motor cortex likely conveys the timing of syllables (that are produced at theta rate) during speaking to the temporal cortex. Indeed, during listening, when the motor cortex does not produce these syllable patterns, this connectivity is reduced. In contrast, beta band signals from the motor cortex to the LSTG are similarly strong in both conditions. These signals have previously been linked to the process of updating sensory predictions and have been observed during various auditory tasks [63,64,69,70] including continuous speech perception [71]. Our correlation results (discussed in detail in the following section) also suggest that top-down beta activity is related to prediction maintenance by carrying speech-related information to the auditory areas. For gamma frequencies, DAI reverses sign to become negative, indicating the dominance of feedforward signals from STG to motor cortex in this frequency band (Fig 7D). Both feedforward and feedback signals between motor and auditory areas are stronger during speaking compared to listening, indicating the stronger coupling of both systems when speech is self-generated. This is consistent with the predictive model since self-generated speech leads to estimations of predicted sensory input that are communicated to auditory areas through low-frequency channels. In turn, the auditory input is used to update these estimations through feedforward signals in the high gamma band. However, since we have not controlled or modified our stimulus material in a way that leads to specific changes in predictions, we cannot assign more precise computational roles to our reported connectivities.

A very similar connectivity pattern indicating feedforward signalling at gamma frequencies and feedback signals at low frequencies is evident for the connection from left parietal cortex to LSTG. However, compared to the motor cortex, top-down effects use a different low-frequency band, namely, alpha. Importantly, alpha rhythms in the auditory system have been linked to attentional control [72,73] and might serve as a multidimensional filter of sensory inputs across space, time, and frequency dimensions [74]. Our connectivity results suggest that these filters in early auditory areas are controlled and modulated by the parietal cortex with its unique anatomical and functional location at the interface between auditory and articulatory speech areas. Indeed, parietal areas are known to support sensorimotor integration [75,76], code-predicted sensory consequences during speaking, and provide top-down signals to early auditory areas [77]. The stronger top-down effects during speaking might indicate the selective inhibition of auditory signals that are predicted by the internal predictive coding model. In fact, this "sensory attenuation" (see also subsection below) has been shown to be reflected in alpha rhythms [78] in accordance with our connectivity results.

Left inferior frontal gyrus (IFG) is crucial for language comprehension and production and contributes to the formation of a predictive model of language [79]. In a TMS study, Ishkhanyan and colleagues targeted anterior and posterior parts of left IFG during an adjective-noun production task and demonstrated the importance of this area for grammatically and syntactically correct speech production [79]. More recently, Castellucci and colleagues demonstrated the major role of IFG in speech planning and preparation [80]. Saur and colleagues reported an anatomical connectivity between left IFG and auditory cortex [81,82]. Similar to motor and parietal cortex, our results show that during speech production, IFG receives feedforward signals from the LSTG in the high gamma band. Yet, another distinct frequency channel is used for feedback signals, namely, the delta band (S3 Fig). The importance of low-frequency top-down signals from left IFG to early auditory areas in the delta band has previously been shown for listening [12,83]. It should be noted that with DAI, we are contrasting the 2 directed connectivities between 2 areas (see Methods section) that makes our analysis specifically sensitive to asymmetries (i.e., one direction being significantly stronger than the opposite direction) and not directly comparable to studies investigating individual directions (e.g., as Gross and colleagues [12]).

In summary, our connectivity results are in line with the theory of predictive coding and suggest the essential role of directional frequency-specific modulation in cortical information processing for speech production and perception. According to the predictive coding framework, predictions are updated by the linear accumulation of prediction errors over time that leads to a slower changing of prediction in comparison to prediction error [17,62]. Thus, prediction error signals need to utilise a higher frequency channel for communication than prediction signals that is in agreement with our findings.

## Sensory attenuation in speech production

In contrast to externally generated sounds, self-generated sounds exhibit smaller cortical responses in the auditory cortex because their sensory consequences can be accurately predicted (known as sensory attenuation) [84]. This is also the case for speech: Previous studies have shown that self-produced speech results in reduced auditory-evoked brain activity compared to passive listening [4–6]. We extend these findings by showing that speech-brain coupling is significantly weaker during speaking than during listening, especially at low frequencies. This is consistent with predictions derived from the sensory attenuation literature and the predictive speech model. According to this model, motor commands for speech

production are not only sent to muscles controlling articulators but also communicated to cortical areas that infer to be expected auditory input. Therefore, during speaking, expected sensory input will be attenuated in early auditory areas leading to a reduced coupling of auditory activity to the (self-generated) speech envelope.

In our previous study, we found that neural responses to self-produced stimuli are modulated by beta-band rhythms in the motor cortex [69]. Additionally, we reported significant directional couplings in the beta range originating from motor cortices towards bilateral auditory areas. In the current study, we also tested whether top-down connectivity directly affected the cortical tracking of speech signals. Interestingly, we found a negative correlation between theta-range speech-brain coupling in the left auditory area and top-down beta connectivity originating from motor areas. This finding is consistent with prior studies showing the role of beta activity in updating predictions about the sensory consequences of the upcoming movements [63,85]. In an MEG study, Morillon and colleagues also reported beta top-down connectivity from motor to auditory cortices is directly linked to temporal predictions [70]. Interestingly, our correlation results showed that attenuation of speech-brain coupling, derived from top-down audio-motor connectivity in the low beta range (12 to 20 Hz), primarily occurs in the low theta band (4 to 6 Hz). These results show the important role of low beta rhythms in interactions between motor and auditory areas in speech production that are in line with previous study introducing low beta activity as a central coordinator of temporal predictions for auditory entrainment [86]. On the other hand, according to previous studies, the maximum speech modulations fall in low theta band (around 4 Hz) [10,87]. Therefore, the decrease in low theta speech-brain coupling and its negative correlation with top-down low beta connectivity support our hypothesis regarding reflecting sensory attenuation phenomenon.

In summary, using multivariate analysis, we reveal the similarities and differences of speech tracking between listening and speaking across the cortex. We show that during speaking, brain activity is most strongly aligned to the speech envelope in auditory and motor areas. Our time-resolved MEG results further demonstrate that this alignment precedes the speech envelope in frontal and motor areas and follows the speech envelope in auditory areas. Indeed, in frontal and motor areas, this speech-brain coupling is stronger for speaking than listening while the opposite is true for auditory areas. Detecting a negative lag coupling beyond the primary motor cortex could potentially support the role of predictive processing models in predicting speech programmes and sensory consequences. More importantly, our directed connectivity analysis suggests that feedforward signals are mostly communicated in the gamma frequency band while feedback signals use low-frequency channels with frequencies depending on the cortical origin of these signals. These findings are consistent with the "predictive coding" framework that would interpret the top-down signals as predictions and bottom-up signals as prediction errors. This communication is also modulated by tasks such that feedback from the frontal-motor cortex is stronger during speaking than listening. Additionally, the negative correlation between top-down beta connectivity originating from the motor areas and theta-range speech-brain coupling in the left auditory area supports the role of beta activity in predicting the sensory consequences of self-generated speech.

## Methods

### Ethics statement

The study was approved by the local ethics committee and conducted in accordance with the Declaration of Helsinki (Approval number: 2018-066-f-S).

## Participants

We recruited 30 native German-speaking participants (15 males, mean age 25.1 ± 2.8 years, range 20 to 32 years) from a local participant pool. Prior written informed consent was obtained before the measurement and participants received monetary compensation after the experiment.

## Recording

MEG, EMG, and speech signals were recorded simultaneously. A 275 whole-head sensor system (OMEGA 275, VSM Medtech, Vancouver, Canada) was used for all of the recordings with a sampling frequency of 1,200 Hz, except the speech recording which had a sampling rate of 44.1 kHz. Audio data was captured with a microphone, which was placed at a distance of 155 cm from the participants' mouth, in order not to cause any artefacts through the microphone itself. Three pairs of EMG surface electrodes were placed after tactile inspection to find the correct location to capture muscle activity from the m. genioglossus, m. orbicularis oris, and m. zygomaticus major (for exact location, see Fig 1 in Abbasi and colleagues [42]). One pair of electrodes was used for each muscle with about 1 cm between electrodes. A low-pass online filter with a 300 Hz cutoff was applied to the recorded MEG and EMG data.

## Paradigm

Participants were asked to sit relaxed while performing the given tasks and to keep their eyes focused on a white fixation cross. This study consisted of 3 separate recordings: (i) speech production; (ii) speech production while perception was masked; and (iii) speech perception. For the speech production recording, there were 7 trials for overt speech. Each trial consisted of a 60-s time period in which participants answered a given question such as "What does a typical weekend look like for you?". A colour change of the fixation cross from white to blue indicated the beginning of the time period in which participants should speak and the end was marked by a colour change back to white. In the second recording, participants were asked to perform the same task as in the first recording while they heard white noise, leaving them unable to hear their own voice. The questions were different to the prior recording in order to prevent repetition and prefabricated answers. To keep emotional answers out of the way, questions covering neutral topics were chosen (full list of questions: S1 Table).

In the third recording session, participants listened to audio recordings of their own voice, which were collected in the first and second recordings. For this paper, only conditions (i) and (iii) were used.

## Preprocessing and data analysis

Prior to data analysis, MEG data were visually inspected. No jump artefacts or bad channels were detected. A discrete Fourier transform (DFT) filter was applied to eliminate 50 Hz line noise from the continuous MEG and EMG data. Moreover, EMG data was high-pass–filtered at 20 Hz and rectified. Continuous head position and rotation were extracted from the fiducial coils placed at anatomical landmarks (nasion, left, and right ear canals). The wideband amplitude envelope of the speech signal was computed using the method presented in [88]. Nine logarithmically spaced frequency bands between 100 and 10,000 Hz were constructed by band-pass filtering (third-order, Butterworth filters). Then, we computed the amplitude envelope for each frequency band as the absolute value of the Hilbert transform and downsampled them to 1,200 Hz. Finally, we averaged them across bands and used the computed wideband amplitude envelope for all further analysis.

MEG, EMG, speech envelope, and head movement signals were downsampled to 256 Hz and were segmented to 60-s trials. In the preprocessing and data analysis steps, custom-made scripts in Matlab R2020 (The Mathworks, Natick, Massachusetts, United States of America) in combination with the Matlab-based FieldTrip toolbox [89] were used in accord with current MEG guidelines [90].

### Artefact rejection

For removing the speech-related artefacts, we used the pipeline presented in our recently published study [42]. In a nutshell, the artefact rejection comprises 4 major steps: (i) head movement-related artefact was initially reduced by incorporating the head position time series into the general linear model (GLM) using regression analysis [91]. (ii) To further remove the residual artefact, SVD was used to estimate the spatial subspace (components) containing the speech-related artefact from the MEG data. (iii) Artefactual components were detected via visual inspections and MI analysis and then removed from the single-trial data [92]. (iv) Finally, all remaining components were back-transformed to the sensor level.

### Source localization

For source localisation, we aligned individual T1-weighted anatomical MRI scans with the digitised head shapes using the iterative closest point algorithm. Then, we segmented the MRI scans and generated single-shell volume conductor models [93] and used this to create forward models. For group analyses, individual MRIs were linearly transformed to an MNI template provided by Fieldtrip. Next, the linearly constrained minimum variance (LCMV) algorithm was used to compute time series for each voxel on a 5-mm grid. The time series were extracted for each dipole orientation, resulting in 3 time series per voxel. The reduced version of the HCP brain atlas was applied on the source space time series in order to reduce the dimensionality of the data, resulting in 230 parcels [45]. Finally, we extracted the first 3 components of an SVD of time series from all dipoles in this parcel, explaining most of the variance.

### Mutual information

For each parcel, the 3 SVD components were subjected to multitaper spectral analysis with +/− 2 Hz spectral smoothing on 2-s long windows with 50% overlap. We then estimated MI using Gaussian Copula MI between the speech envelope and complex-valued spectral estimates of all 3 time series [46]. Note that due to the multivariate nature of the computation, this resulted in a single MI value for the parcel that combines information across all 3 time series. This computation was repeated for each parcel and 52 delays between speech envelope and neural activity ranging (equally spaced) from −1 to 1 s. For each delay, the shifting of the speech envelope with respect to the MEG signal was performed before computing the multitaper spectral estimate. In addition, we computed a surrogate distribution reflecting the expected MI values in the absence of true synchronisation by computing MI for 500 random delays between speech envelope and source activity (only delays longer than 3 s were used). We implemented shifting with the matlab function circshift.m with wrapping around the edge of the time series.

### Delay estimation

To estimate the optimal lag of speech-brain coupling during speaking (Fig 3), we used the t-values resulting from the comparison of speaking against the 95th percentile of surrogate data. The t-values had been computed across lags (−1 s to 1 s) and frequencies (1 to 10 Hz) and *p*-

values had been FDR-corrected for multiple comparisons across lags and frequencies. First, we averaged the t-values across brain areas yielding the global speech-brain coupling signature across lags and frequencies. Next, we computed for each brain area the relative change of t-values compared to the global signature resulting in a matrix of relative changes across lag and frequency for each brain area. Finally, we averaged this matrix across frequencies and identified the lag where this averaged relative change was maximal. Please note that this procedure only captures the lag corresponding to the global peak across lags even if multiple peaks are present.

## Connectivity analysis

We performed connectivity analysis by using a multivariate nonparametric Granger causality approach (mCG) [49]. We computed the mCG to determine the directionality of functional coupling between STG (LA5 parcel in HCP atlas) and each cortical area, in pairwise steps, during speech production and perception. Initially, the source signals were divided into trials of 4 s, with 500 ms overlap. We used the fast Fourier transform in combination with multitapers (2 Hz smoothing) to compute the cross-spectral density (CSD) matrix of the trials. Next, using a blockwise approach, we considered the first 3 SVD components of each parcel as a block and estimated the connectivity between STG and other parcels. Finally, we computed the directed influence asymmetry index (DAI) defined by [15] as

$$DAI = \frac{[mGC(parcel \rightarrow STG) - mGC(STG \rightarrow parcel)]}{[mGC(parcel \rightarrow STG) + mGC(STG \rightarrow parcel)]}.$$

Therefore, a positive DAI for a given frequency indicates that the selected parcel conveys feedforward influences to STG in this frequency, and a negative DAI indicates feedback influences. Note that for the connectivity analysis, we used MEG data with 1,200 Hz sampling rate without downsampling.

## Statistical analysis

We determined significant normalised MI values as well as connectivity patterns (DAI values) in both speaking and listening conditions using nonparametric cluster-based permutation tests [94]. First, we estimated the statistical contrast of speech-brain coupling during speaking compared to the 95th percentile of MI for each parcel and participant. Second, the normalised MI values in the speaking condition were contrasted with MI values in the listening condition at the group level. The statistical analysis was conducted for each frequency (in the range 1 to 10 Hz) and delay (in the range −1 to 1 s with 52 equally spaced delays) using a dependent-samples $t$ test. We used a cluster-based correction to account for multiple comparisons across frequencies, delays, and parcels. We performed 5,000 permutations and set the critical alpha value at 0.05. We performed the same procedures in order to detect significant connectivities between STG and other cortical parcels in both speaking and listening conditions.

## Correlation analysis

We assessed whether there is a relationship between our connectivity results from all the parcels to LSTG and speech-STG coupling using nonparametric cluster-based permutation tests. First, we estimated the top-down connectivity values for each parcel and frequency band. Next, we computed speech-STG couplings for each frequency band. The correlation analysis was conducted for each frequency band and parcel using the Pearson method implemented in the *ft_statfun_correlationT* function in Fieldtrip. We used cluster-based correction to account

for multiple comparisons across parcels. Our analysis was repeated for different frequency bands. Therefore, our results are not corrected across frequencies. We performed 5,000 permutations and set the critical alpha value at 0.05.

## Supporting information

**S1 Table. Full list of questions participants had to answer in the first (speech production) and second recording (speech production with masked perception).**
(DOCX)

**S1 Fig. Results of speech brain coupling for the listening condition.** Cortical map represents groups statistics of multivariate MI between speech envelope and brain activity in each parcel compared to 95th percentile of surrogate data. Colour code represents the sum of all significant T-values across delay and frequency (FDR-corrected across delays (−1 to 1 s), frequencies (1 to 10 Hz), and parcels). The data underlying this figure can be found in https://osf.io/9fq47/
.
(DOCX)

**S2 Fig. Upper panel: Statistical map of fMRI meta-analysis from neurosynth.org using the search term "speech production".** Results were spatially interpolated to the HCP atlas used in our study. Colour codes correspond to z-values from a uniformity test as documented on neurosynth.org. Lower panel: The relative difference of correlations (r1-r2)/(r1+r2) across delays. r1 is the correlation between our speech production network and the neurosynth speech production network and r2 is the correlation between our speech production network and the neurosynth speech perception network. The data underlying this figure can be found in https://osf.io/9fq47/.
(DOCX)

**S3 Fig. Speech-brain coupling lateralisation for listening condition.** The data underlying this figure can be found in https://osf.io/9fq47/.
(DOCX)

**S4 Fig. Statistical comparison of speech-brain coupling in speaking versus listening at delta (2 Hz) and theta (5 Hz) for different delays ([−600 600] ms).** Colour codes t-values. The data underlying this figure can be found in https://osf.io/9fq47/.
(DOCX)

**S5 Fig. Left STG area.** The highlighted black parcel depicts the L_A5 parcel of the HCP atlas representing the LSTG area. The data underlying this figure can be found in https://osf.io/9fq47/.
(DOCX)

**S6 Fig.** Significant connectivity between STG and other cortical parcels during speech production (a) and listening (b) in different frequency bands. A cluster-based permutation test was used to detect significant connectivity patterns ($p < 0.05$). Colour codes t-values. Purple colour represents the flow of information from STG to other cortical parcels and the green colour represents the opposite direction. The data underlying this figure can be found in https://osf.io/9fq47/.
(DOCX)

**S7 Fig. Comparison of connectivity patterns between speaking and listening conditions in different frequency bands.** A cluster-based permutation test was used to detect significant connectivity differences from all the cortical parcels to STG between speaking and listening

conditions ($p < 0.05$). Colour codes t-values. Please note that since we compared asymmetry indices (DAI) between 2 conditions, interpreting the directionality from this cortical plot is challenging. For a better understanding of these statistical maps, please refer to the spectrally resolved DAI between STG and 4 ROIs in Fig 7D as well as S3 Fig. The data underlying this figure can be found in https://osf.io/9fq47/.
(DOCX)

**S8 Fig. Correlation analysis between top-down GC and MI.** Speech-STG coupling in theta range (positively lagged: 130 ms) is negatively correlated with the top-down delta (a) as well as theta (b) connectivity from mainly left occipital and parietal areas. The data underlying this figure can be found in https://osf.io/9fq47/.
(DOCX)

**S9 Fig. Frequency-specific power comparison between speaking and listening condition.** (**a**) The statistical contrast of frequency-specific power between speaking and listening. Results of dependent-samples *t* test were FDR corrected across parcels and frequencies (1–100 Hz). Colour codes sum of t-values across all significant frequencies. Note that we use a one-tailed test (speaking < listening) to ensure that results are not contaminated by residual speech arte-facts. (**b**) Power spectral density averaged across all significant parcels for speaking (blue) and listening (red) conditions. The grey panel depicts the frequency range 10–30 Hz. The data underlying this figure can be found in https://osf.io/9fq47/.
(DOCX)

## Author Contributions

**Conceptualization:** Joachim Gross.

**Data curation:** Nadine Steingräber.

**Formal analysis:** Omid Abbasi, Nadine Steingräber, Joachim Gross.

**Funding acquisition:** Joachim Gross.

**Investigation:** Joachim Gross.

**Methodology:** Omid Abbasi, Joachim Gross.

**Project administration:** Joachim Gross.

**Validation:** Omid Abbasi.

**Visualization:** Omid Abbasi, Nikos Chalas, Daniel S. Kluger.

**Writing – original draft:** Omid Abbasi, Joachim Gross.

**Writing – review & editing:** Omid Abbasi, Nadine Steingräber, Nikos Chalas, Daniel S. Klu-ger, Joachim Gross.

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
