## [Editor Report · Decision Letter 0]

9 Feb 2023

Dear Dr Abbasi, 

Thank you for submitting your manuscript entitled "Frequency-specific brain networks in continuous speaking and listening" for consideration as a Research Article by PLOS Biology and using our Portable Peer Review system. I have discussed your manuscript with the in-house editorial team and with an Academic Editor with relevant expertise and am writing to let you know we feel the study has potential for PLOS Biology. Your revision seems quite responsive to the questions and concerns that had been raised in the prior rounds of peer review at Nature Communications. We would therefore like to have the reviewers take a look at this new revision before making a final call. I have reached out to Nature Communications to see if we can obtain these reviewer identities. If we are able to get these, we will go back to the same reviewers. If we are unable to obtain these identities, we will approach a few new reviewers but will ask them to specifically comment on the revision in light of the reviews you've already received.

Before we can send your manuscript to reviewers, we will also need you to complete your submission by providing the metadata that is required for full assessment. To this end, please login to Editorial Manager where you will find the paper in the 'Submissions Needing Revisions' folder on your homepage. Please click 'Revise Submission' from the Action Links and complete all additional questions in the submission questionnaire.

Once your full submission is complete, your paper will undergo a series of checks in preparation for peer review. After your manuscript has passed the checks it will be sent out for review. To provide the metadata for your submission, please Login to Editorial Manager (https://www.editorialmanager.com/pbiology) within two working days, i.e. by Feb 11 2023 11:59PM.

Feel free to email us at plosbiology@plos.org if you have any queries relating to your submission, or to reach out to me directly at kdickson@plos.org.

Kind regards,

Kris

Kris Dickson, Ph.D., (she/her)

Neurosciences Senior Editor/Section Manager

PLOS Biology

kdickson@plos.org

---

## [Decision Letter · Decision Letter 1]

12 May 2023

Dear Dr Abbasi,

Thank you for your patience while we considered your revised manuscript "Frequency-specific brain networks in continuous speaking and listening" for publication as a Research Article at PLOS Biology. This revised version of your manuscript has been evaluated by the PLOS Biology editors, the Academic Editor and two of the original Nature Communications reviewers. Unfortunately we were not able to obtain the identity of reviewer #1, but the Academic Editor has assessed your responses to that reviewer (as has reviewer #2). Please accept my apologies for the extreme delay incurred earlier in the process.

Based on the reviews and our Academic Editor's assessment of your revision, we are likely to accept this manuscript for publication, provided you satisfactorily address the remaining points raised by the reviewers and the following data and other policy-related requests.

IMPORTANT - Please attend to the following:

a) Please could you make your Title more informative and accessible, preferably including an active verb? Based on your Abstract, I'd suggest something like "Brain networks involved in continuous speaking and listening are partially shared but dissociated in space, time and frequency," but you will likely be able to fashion something more accurate.

b) Please provide a blurb, according to the instructions in the submission form.

c) Please address the remaining concerns from the two reviewers.

c) Please address my Data Policy requests below; specifically, we need you to supply the numerical values underlying Figs 2, 3AB, 4, 5AB, 6ABCD, 7ABCD, 8AB, 9ABCDE, S1, S2AB, S3, S4AB, S5, S6AB, S7, S8AB, S9, either as a supplementary data file or as a permanent DOI’d deposition. We note that you have already deposited MatLab code and a small subset of the data in OSF; I will need to see the full deposition before accepting the paper for publication. I understand that the raw data are protected by privacy laws, and are therefore exempt from PLOS’ data policy.

d) Please cite the location of the data clearly in all relevant main and supplementary Figure legends, e.g. “The data underlying this Figure can be found in S1 Data” or “The data underlying this Figure can be found in https://osf.io/XXXXX"

e) We note that you state that "The study was approved by the local ethics committee" - please could you also include an approval number?

We expect to receive your revised manuscript within two weeks. 

*Published Peer Review History*

*Press*

Sincerely,

Roli Roberts

Roland Roberts, PhD

Senior Editor,

rroberts@plos.org,

PLOS Biology

DATA POLICY:

Regardless of the method selected, please ensure that you provide the individual numerical values that underlie the summary data displayed in the following figure panels as they are essential for readers to assess your analysis and to reproduce it: Figs 2, 3AB, 4, 5AB, 6ABCD, 7ABCD, 8AB, 9ABCDE, S1, S2AB, S3, S4AB, S5, S6AB, S7, S8AB, S9. NOTE: the numerical data provided should include all replicates AND the way in which the plotted mean and errors were derived (it should not present only the mean/average values).

DATA NOT SHOWN?

REVIEWERS' COMMENTS:

Reviewer #1:

[identifies himself as Benjamin Morillon]

I reviewed a previous version of the manuscript for another journal. My current review addresses the comments raised by all the previous reviewers. As I previously wrote, this work is innovative, methodologically and conceptually, the methods are sound, and the results are important to further our understanding of the brain as being a dynamical system.

Concerning the previous concerns of Reviewer #1, I tend to agree with their, and appreciate that the authors have deemphasized references to predictive coding from their manuscript. 

However, I would suggest removing any mention of predictive coding in the abstract (where it is present in two instances). Indeed, none of the results investigate predictive processing per se, as is for instance done with the estimation of surprisal or entropy variables that are then correlated to brain dynamics and connectivity patterns. Here predictive coding is only one of the possible interpretations of their result (and not the most straightforward).

The authors also added a fifth question in the introduction, but again I would use caution regarding this point (and actually, I suggest to simply remove it), for a specific set of related reasons:

- First, beta (∼20 Hz) activity is the default oscillatory mode of the motor system.

- Second, the framework championed by A. Bastos and colleagues (that the authors rely on here) has recently been heavily put into question, notably by the work of M. Vinck and colleagues (see, e.g. Schneider et al., Neuron 2021, or Vinck Neuron 2023 - but also by the group of C.E. Schroeder or A. Thiele). They basically show that inter-areal directed coherence can be predicted by the dominant power amongst the two explored regions.

- Third, regrouping these two information results in the hypothesis that beta activity will be necessarily be directed from motor to auditory regions. Imho, this reflects the most parsimonious hypothesis, and there is no need to rely on predictive coding assumptions of dedicated frequency channels.

- Finally, as I said before, the authors entirely rely on the proposal by Bastos et al. (which was actually never empirically validated!) but do not show any evidence that some predictions or prediction error signals are transmitted between motor and auditory regions in the beta band. It is simply an interpretation of their results that they can of course mention in the discussion. But I would use caution (and mention alternative interpretations too, such as the one described above, for instance).

Regarding the negative-lag result, I would also first emphasize the most accepted interpretations, notably that the motor cortex is involved in motor preparation, then in the processing of auditory feedback (see, e.g. Ozker…Flinker, PloS Biol 2022 for a relevant example), and then in predictive processing.

I have no further comments and congratulate the authors for their excellent work.

Reviewer #3:

The study's main contribution is a comprehensive description of the brain tracking of the speech envelope during continuous speaking and listening. Moreover, it also describes connectivity patterns that could be interpreted in the light of the predictive coding framework.

Since a substantial part is dedicated to the understanding of the neural substrate of speaking, the inclusion of relevant related bibliography could strengthen the arguments and provide additional context to the findings:

Pérez, A. et al. Timing of brain entrainment to the speech envelope during speaking, listening and self-listening. Cognition (2022).

Ozker M. et al. A cortical network processes auditory error signals during human speech production to maintain fluency. Plos Biology (2022).

Page 5 of the document: the narrative ends abruptly with the disconnected paragraph "Recently, we characterised …". 

To improve clarity, the captions of Figures 3 and 4 should include "during speaking".

---

## [Editor Report · Decision Letter 2]

31 May 2023

Dear Dr Abbasi,

Thank you for the submission of your revised Research Article "Spatio-temporal dynamics characterise spectral connectivity profiles of continuous speaking and listening" for publication in PLOS Biology. On behalf of my colleagues and the Academic Editor, David Poeppel, I'm pleased to say that we can in principle accept your manuscript for publication, provided you address any remaining formatting and reporting issues. These will be detailed in an email you should receive within 2-3 business days from our colleagues in the journal operations team; no action is required from you until then. Please note that we will not be able to formally accept your manuscript and schedule it for publication until you have completed any requested changes.

Sincerely, 

Roli Roberts

Senior Editor

PLOS Biology

rroberts@plos.org